# Intensity and Dynamics of Anti-SARS-CoV-2 Immune Responses after BNT162b2 mRNA Vaccination: Implications for Public Health Vaccination Strategies

**DOI:** 10.3390/vaccines10020316

**Published:** 2022-02-17

**Authors:** Matthaios Speletas, Ioanna Voulgaridi, Styliani Sarrou, Aikaterini Dadouli, Varvara A. Mouchtouri, Dimitrios J. Nikoulis, Maria Tsakona, Maria A. Kyritsi, Athanasia-Marina Peristeri, Ioanna Avakian, Asimina Nasika, Paraskevi C. Fragkou, Charalampos D. Moschopoulos, Stamatia Zoubouneli, Ilias Onoufriadis, Lemonia Anagnostopoulos, Alexia Matziri, Georgia Papadamou, Aikaterini Theodoridou, Sotirios Tsiodras, Christos Hadjichristodoulou

**Affiliations:** 1Department of Immunology & Histocompatibility, Faculty of Medicine, School of Health Sciences, University of Thessaly, 41500 Larissa, Greece; maspel@med.uth.gr (M.S.); ssarrou@uth.gr (S.S.); atperist@uth.gr (A.-M.P.); ionoufriadis@uth.gr (I.O.); katetheod41222@gmail.com (A.T.); 2Laboratory of Hygiene and Epidemiology, Faculty of Medicine, University of Thessaly, 41222 Larissa, Greece; ioavoulg@uth.gr (I.V.); adadouli@uth.gr (A.D.); mouchtourib@med.uth.gr (V.A.M.); dnikoulis@uth.gr (D.J.N.); mkiritsi@med.uth.gr (M.A.K.); joavakian@uth.gr (I.A.); anasika@uth.gr (A.N.); lanagnost@uth.gr (L.A.); alexmatz@uth.gr (A.M.); 3Fourth Department of Internal Medicine, School of Medicine, Attikon University Hospital, National and Kapodistrian University of Athens, 12462 Athens, Greece; mariatsakwna1@gmail.com (M.T.); pcfragkou@med.uoa.gr (P.C.F.); chamos@med.uoa.gr (C.D.M.); tsiodras@med.uoa.gr (S.T.); 4Emergency Department, University Hospital of Larissa, 41500 Larissa, Greece; szoumpounelli@uth.gr (S.Z.); georgia.papadamou@yahoo.com (G.P.)

**Keywords:** COVID-19, vaccination, IgG, IgA, antibody responses, T cell responses

## Abstract

The aim of our study was to investigate the immunogenicity of the BNT162b2 vaccination according to the age and medical status of vaccinated individuals. A total of 511 individuals were enrolled (median age: 54.0 years, range: 19–105); 509 of these individuals (99.6%) received two doses of BNT162b2 at an interval of 21 days. IgG and IgA responses were evaluated on days 21, 42, 90, and 180 after the first dose with chemiluminescent microparticle and ELISA assays. The cell-mediated immune responses were assessed by an automated interferon-gamma release assay. We demonstrated positive antibody responses after vaccination for the majority of enrolled participants, although waning of IgG and IgA titers was also observed over time. We further observed that the intensity of humoral responses was positively correlated with increased age and prior COVID-19 infection (either before or after the first vaccination). Moreover, we found that only a medical history of autoimmune disease could affect the intensity of IgA and IgG responses (3 weeks after the primary and secondary immunization, respectively), while development of systemic adverse reactions after the second vaccination dose was significantly associated with the height of IgG responses. Finally, we identified a clear correlation between humoral and cellular responses, suggesting that the study of cellular responses is not required as a routine laboratory test after vaccination. Our results provide useful information about the immunogenicity of COVID-19 vaccination with significant implications for public health vaccination strategies.

## 1. Introduction

The coronavirus disease of 2019 (COVID-19), due to severe acute respiratory syndrome coronavirus 2 (SARS-CoV-2), has dramatically affected our world, resulting in a substantial loss of life and overwhelming health care systems [1,2]. Consequently, the development of a safe and effective COVID-19 vaccine rapidly became a global priority [2]. The BNT162b2 mRNA vaccine developed by BioNTech and Pfizer [3] was the first vaccine used in Greece against COVID-19 beginning in December 2020 following emergency use authorization by the European Medicines Agency (EMA).

The BNT162b2 mRNA vaccine is a nucleoside-modified RNA formulated in lipid nanoparticles that encodes the spike (S) glycoprotein of SARS-CoV-2 [4,5]. As reported, the S protein contains the receptor-binding domain (RBD), which binds to the angiotensin-converting enzyme 2 (ACE2) located in the target cell membrane, gaining access to the cells. The RBD is the main target of neutralizing antibodies against SARS-CoV-2 following a natural infection. Anti-nucleoprotein (N) antibodies can also develop; however, these antibodies are unable to neutralize the virus in humans [2,6].

The recommended vaccination schedule of BNT162b2 is an intramuscular administration of two 30 μg doses with an interval of 21 days between doses [4,5]. The mRNA is translated into SARS-CoV-2 S protein, which is transiently expressed on the surface of host cells; this expression induces humoral and cellular immune responses conferring protection against COVID-19 [4,5]. Several studies have demonstrated the safety and immunogenicity of BNT162b2 vaccination [4,6,7]. However, immunogenicity varies and may not only be dependent on age and history of previous SARS-CoV-2 infection, as has been evaluated in previous studies [6,7]. Other factors which remain unknown could affect the intensity and duration of protection against COVID-19 (e.g., time elapsed from primary series, co-existence of other acute or chronic medical conditions, or use of specific medications).

Therefore, the aim of our study was to establish immunogenicity in the context of intensity and duration of immune responses to BNT162b2 vaccination in Greece according to the age and medical status of vaccinated individuals. Our results provided useful information about the immunogenicity of COVID-19 vaccination with significant implications for public health vaccination strategies.

## 2. Materials and Methods

### 2.1. Subjects

A total of 511 individuals were enrolled in the study (male/female: 190/321, median age: 54.0 years, range: 19–105) and received 2 doses of the BNT162b2 mRNA vaccine (Pfizer-BioNTech, New York, NY, USA) with an interval of 21 days between doses. Sixty individuals (11.7%) displayed a positive medical history of COVID-19 (defined by a positive polymerase chain reaction result on a nasopharyngeal swab) at least 3 months prior to vaccination and were vaccinated against SARS-CoV-2 according to the Greek Ministry of Health guidelines. The only exclusion criterion was the presence of primary antibody deficiency (PAD) syndrome, since the kinetics of antibody responses and cellular immunity after vaccination in PAD patients was the subject of another study. Serum and whole blood samples were collected at: (a) day 21, directly before the second vaccination; (b) day 42; (c) day 90; and (d) day 180 after the first vaccination. All individuals received their first dose of the BNT162b2 vaccine between 31 December 2020 and 20 June 2021. An estimation of antibody kinetics was performed for all individuals at least twice (464 on day 21, 475 on day 42, 407 on day 90, and 322 on day 180). An overview of enrolled individuals’ demographic data and medical history is presented in Table 1.

All enrolled individuals or relatives of individuals with insomnia or mental disorders provided informed consent, including details of their medical histories (Table 1). The study was conducted according to the principles of the Helsinki Declaration and was approved by the ethical committee of the Faculty of Medicine, University of Thessaly (No. 2116).

### 2.2. Laboratory Analyses

To assess humoral responses to vaccination, IgG and IgA antibodies targeting the SARS-CoV-2 spike (S1) protein were detected in serum with commercially available kits. The presence of anti-SARS-CoV-2 anti-S IgG antibodies was determined using an ABBOTT SARS-CoV-2 IgG II assay, a chemiluminescent microparticle immunoassay (CMIA), with an ARCHITECT i2000SR immunoassay analyzer (Abbott, Abbott Park, IL, USA). The measurement of anti-SARS-CoV-2 anti-S IgA antibodies was estimated using a SERION ELISA agile SARS-CoV-2 IgA ESR400A kit (Serion, Würzburg, Germany) according to the manufacturer instructions. Moreover, anti-nucleocapsid (anti-N) anti-SARS-CoV-2 IgG antibodies, determined as described by an ABBOTT SARS-CoV-2 IgG assay [8], were also used as markers of SARS-CoV-2 infection before and after vaccination.

To assess cell-mediated immune responses to vaccination on day 180 following the first dose, SARS-CoV-2-specific IFN-gamma release in whole blood was measured in select individuals (10 with positive humoral responses and 11 with negative humoral responses); a SARS-CoV-2 IGRA stimulation tube set from Euroimmun (Lübeck, Germany) was used according to the manufacturer instructions. The whole blood was incubated for 20 h at 37 °C with: (a) no activating components for immune cells; (b) components of the S1 domain of the SARS-CoV-2 spike protein; or (c) a mitogen causing an unspecific interferon-gamma secretion. Subsequently, IFN-gamma was quantified with an ELISA EQ 6941-9601 kit (Euroimmun). A cut-off value of 100 mIU/mL was used to discriminate negative cell-mediated immune responses to S-protein of SARS-CoV-2 from borderline (100–200 mIU/mL) and positive (>200 mIU/mL) responses. All samples displayed a positive response to mitogen.

### 2.3. Statistical Analysis

Categorical variables were described with the use of frequency and relative frequency. Continuous variables were described with the use of median and interquartile range (IQR). Antibody levels were compared by age group (<30, 30–39, 40–49, 50–59, 60–69, 70–79, and >80 years), as well as by infected versus uninfected individuals, using non-parametric *t* tests (Mann-Whitney U and Kruskal-Wallis H tests). Analysis of continuous variables was conducted using the Mann-Whitney U test and correlations were made using Spearman’s rank correlation coefficient. Data were checked for deviation from normal distribution using the Shapiro-Wilk normality test. Kaplan-Meier curves were used to estimate the probability of antibody loss at different time points, and a log-rank test was employed to assess the differences between covariates. The event of interest was a negative antibody test. Multivariate analysis was performed using multiple regression and Cox regression techniques. Multiple regression was used to determine independent predictors of antibody quantities and levels, while Cox regression was used to determine independent predictors of antibody loss. For all analyses, a 5% significance level was set. Analysis was carried out with SPSS (version 25.0) and GraphPad Prism (version 9.2.0) software.

## 3. Results

### 3.1. Safety and Adverse Reactions to BNT162b2 Vaccination

A total of 509 individuals (99.6%) were vaccinated twice according to the initial vaccination schedule. Two individuals did not receive the second dose: a 58 year old female (due to severe facial flushing and electrocardiogram (ECG) changes) and a 95 year old female who refused the scheduled second vaccination without providing a reason.

Detailed information recording the side effects of both vaccination doses was available for 364 individuals; for an additional 102 individuals, it was only known whether or not fever occurred. As presented in Table 2, local side effects were more common after the first dose, while systemic side effects were more common following the second dose. However, in all cases, the intensity and duration of adverse reactions were limited and acceptable, with the exception of one case as mentioned above.

### 3.2. COVID-19 Disease before and after Vaccination in the Study Individuals

As mentioned in the Materials and Methods section, 60 patients reported a previous history of COVID-19, ranging from 3 to 8 months prior to vaccination. Anti-N IgG anti-SARS-CoV-2 antibodies, which are indicative of prior infection, were additionally detected in six individuals at enrollment. These individuals did not report symptomatic COVID-19 infection and most likely experienced asymptomatic infection. Thus, the total number of participants with a previous history of COVID-19 was 66 (12.9%). Moreover, seven individuals in our cohort (1.4%) were infected by SARS-CoV-2 after vaccination, as confirmed by either detection of the virus via RT-PCR on nasopharyngeal swabs or detection of anti-N IgG anti-SARS-CoV-2 antibodies over time. None of these individuals displayed severe disease or required hospitalization, supporting the notion that COVID-19 vaccination is effective protection against severe COVID-19.

Figure 1 presents in detail the dynamics of anti-SARS-CoV-2 anti-S IgG responses of all individuals who were infected post-vaccination. Furthermore, Appendix A details the clinical course and laboratory findings of a participant infected by SARS-CoV-2 post-vaccination who displayed only anosmia for one day.

### 3.3. Intensity and Dynamics of IgG and IgA Responses after BNT162b2 Vaccination

The great majority of vaccinated individuals exhibited a positive antibody response after vaccination, especially those of IgG anti-SARS-CoV-2 origins (Figure 2). In particular, the frequency of individuals with positive IgG and IgA anti-SARS-CoV-2 antibodies were 83.8% and 45.9% on day 21, 97.9% and 86.9% on day 42, and 97.8% and 39.3% on day 90 after the first dosage, respectively. On day 180, 94.7% of individuals displayed positive IgG anti-SARS-CoV2 antibodies, while IgA antibodies were not measured, as most enrolled individuals displayed low levels of IgA, even on day 90 (Figure 2). As presented in Figure 2, the levels of both IgG and IgA anti-SARS-CoV-2 antibodies increased on day 42, and significantly decreased 3 and 6 months after the first vaccination (*p* < 0.001). As mentioned above, most of the vaccinated individuals lost IgA anti-SARS-CoV-2 antibodies 3 months following vaccination and displayed very low levels of IgG antibodies 3 months later (Figure 2).

### 3.4. Correlation of IgG Antibody Responses with Demographic and Clinical Parameters of the Vaccinated Individuals

As presented in detail in Figure 3, Figure 4, Figure 5 and Figure 6 and Table 3 and Table 4, age and history of COVID-19 infection (either before or after vaccination) were the most important factors affecting the intensity and dynamics of IgG anti-SARS-CoV-2 levels after vaccination. In particular, increased age was associated with lower IgG levels over time, a finding that was more profound 6 months after vaccination for individuals over 60 years of age (Figure 3). Moreover, a history of COVID-19, either symptomatic or asymptomatic (confirmed by RT-PCR or the presence of positive anti-N anti-SARS-CoV-2 antibodies), was found to be significantly associated with higher IgG levels over time (Figure 5).

Further analysis revealed that, on day 21 following the first vaccine dose, several factors from patients’ medical histories affected the intensity of the IgG responses (Appendix A). However, when we excluded from analysis the participants with a history of COVID-19 (considering, as mentioned, that history of COVID-19 significantly affected the intensity of IgG antibody responses over time), increased age was the only significant factor affecting IgG levels in multivariate analysis (*p* < 0.001). Similar findings were observed on days 42, 90, and 180 after the first dose, as detailed in Appendix A. Interestingly, only a medical history of autoimmunity affected the intensity of the IgG responses on day 90 in multivariate analysis; patients with autoimmune disorders displayed significantly lower anti-SARS-CoV-2 IgG levels compared to other participants (mean ± SDEV: 4354.1 ± 8787.7 AU/mL vs. 4749.6 ± 7729.0 AU/mL; *p* = 0.029, Appendix A). In this context, a survival multivariate analysis revealed that increased age was the only risk factor that significantly affected the possibility of IgG negativity over time (*p* < 0.001, Appendix A).

Considering the effect of adverse side reactions on the intensity of IgG antibody responses, multivariate analysis revealed that there was no effect on day 21, where only increased age was significant (Appendix A). However, participants displaying fever and myalgias after the second dose exhibited significantly higher IgG levels on day 42 (*p* = 0.004 and *p* = 0.038, respectively; Table 5). Interestingly, this effect was lost on days 90 and 180 post-vaccination (*p* > 0.05, in all cases).

### 3.5. Correlation of IgA Antibody Responses with Demographic and Clinical Parameters of the Vaccinated Individuals

Similar to IgG levels, the anti-SARS-CoV-2 IgA levels post-vaccination were significantly affected by a history of COVID-19 over time (Figure 4). As presented in Figure 3 and Appendix A, age, male gender, and a medical history of autoimmunity significantly affected IgA levels on day 21 after vaccination in multivariate analysis (*p* = 0.001, *p* = 0.019, and *p* = 0.028, respectively). However, contrary to observation of IgG levels, age did not significantly affect IgA levels on day 42 or 3 months following the first vaccination dose (Figure 4). Moreover, no significant differences for gender, autoimmunity, or other comorbidities were observed on days 42 and 90 after vaccination (*p* > 0.05, in all cases). Finally, adverse side effects following the primary or secondary immunization did not affect the intensity of IgA responses in multivariate analysis over time (*p* > 0.05, in all cases). As mentioned above, considering that most enrolled participants lost IgA antibodies on day 90, we did not analyze IgA levels on day 180 post-vaccination.

### 3.6. Correlation of Antibody to Cellular Anti-SARS-CoV-2 Immune Responses in Vaccinated Study Participants

The presence of anti-SARS-CoV-2-reactive T cells was estimated according to the secretion of interferon (IFN) gamma in an ELISA assay after incubation of peripheral blood for 20 h with S peptides from SARS-CoV-2 proteins. All analyzed participants with positive anti-SARS-CoV-2 IgG antibodies had a positive SARS-CoV-2-reactive T cell response in this assay. Conversely, three individuals with negative anti-SARS-CoV-2 IgG antibodies displayed detectable SARS-CoV-2-reactive T cell responses, although their intensities were low. As presented in Figure 7, the magnitude of responses was highly variable, and a good correlate of the response was the height of antibody responses.

## 4. Discussion

Our study provided significant insights into COVID-19 vaccination strategies for individuals of varying ages. We clearly demonstrated positive antibody responses after BNT162b2 vaccination for the great majority of enrolled participants, although a waning of IgG and IgA titers was also observed over time. We observed that the intensity of humoral responses mainly correlated with increased age and prior COVID-19 infection (either before or after first vaccination). Moreover, we found that only a medical history of autoimmune disease could affect the intensity of IgA and IgG responses (3 weeks after the primary and secondary immunization, respectively), while development of systemic adverse reactions after the second vaccination dose was significantly associated with the height of IgG responses. Finally, we identified a clear correlation between humoral and cellular responses, suggesting that the study of cellular responses after vaccination is not needed as a routine laboratory test in order to assess vaccination immunogenicity.

As shown in previous studies [6,7], we observed a strong correlation between the level of humoral anti-SARS-CoV-2 IgG responses and age, with younger participants displaying significantly higher titers of antibodies over time. However, such a finding was not observed in serum IgA levels, which significantly decreased 3 months after vaccination for most participants. Moreover, we did not identify a correlation between IgA levels and participants’ ages. A possible explanation may be the different biology and distribution of immunoglobulins produced after infection or immunization, also considering that, normally, the half-life of human serum IgG is approximately 19 to 21 days, while IgA (which is mainly synthesized by mucosal tissues) has a serum half-life of 5 to 7 days [9]. We cannot exclude the possibility that other factors which remain unknown may affect the height of humoral IgA anti-SARS-CoV-2 responses after vaccination, which could be clarified in future studies.

Our data indicate that the waning pattern of humoral immunity after vaccination differs compared to that of natural immunity after infection. Zeng et al. observed that the titer waning of anti-SARS-CoV-2 IgG antibodies one year after natural infection was more profound in younger convalescent COVID-19 patients (35 years of age or less) compared to older patients [10], which is a completely different pattern than we observed after vaccination. This finding could be attributed to different levels of primary infection severity or duration of virus clearance. Older patients display both more severe disease and longer-term presence of SARS-CoV-2 antigens, which may be accompanied by higher antibody titers for longer time periods, compared to younger patients (who usually present mild or asymptomatic disease) [2,11]. Similar to our results, Ponticelli et al. [12] observed a persistence, but also a waning, of IgG responses 6 months following administration of the BNT162b2 vaccine. Moreover, they demonstrated that IgG responses began to decrease from the second month post-vaccination; however, they did not identify any correlation. The discrepancy with our results may be attributed to the fact that Ponticelli et al. analyzed the immune responses of healthcare workers after vaccination, and this group did not include elderly individuals. This is an advantage of our study, since we evaluated the immune responses after vaccination on a wider spectrum of age.

Similar to previous studies [7,12], we observed that infected participants displayed higher IgG antibody levels after vaccination, further supporting the notion that only one vaccination dose, and not two doses with a 21-day interval, may be adequate for protection from re-infection. However, even for several of these individuals, the titers of IgG levels decreased over time, suggesting that, with the exception of the first dose, the schedule of subsequent vaccinations should be similar to that of uninfected individuals. Moreover, this effect could be hampered by the emergence of newer variants, such as Delta and Omicron [13].

To the best of our knowledge, our study is the largest in the literature analyzing the pattern of IgA immune responses after anti-SARS-CoV-2 vaccination and providing clear evidence that IgA levels display a rapid decline compared to IgG levels. Our results further support the initial studies of Danese et al. and Wisnewski et al., where a rather similar pattern of anti-SARS-CoV-2 IgA serum responses was observed in three and four patients, respectively [14,15]. As mentioned above, we observed that IgA levels were not associated with the age of vaccinated individuals over time but were strongly correlated with a previous history of COVID-19. Currently, several efforts have focused on the development of anti-SARS-CoV-2 immunization through intranasal vaccines, which can trigger mainly mucosal IgA responses to effectively stop respiratory infections [16]. Consequently, the establishment of factors affecting dynamics of IgA both in serum and mucosal tissues should be clarified in further studies.

It is worth noting that the neutralization level after COVID-19 vaccination is positively correlated with protection against COVID-19 [12,17], and several studies report a relationship between the intensity of humoral responses and the height of neutralizing antibody titers [18]. Thus, the extremely mild clinical course of COVID-19 in seven vaccinated participants of our study (who developed adequate humoral responses after vaccination) further supports the notion of COVID-19 vaccination as an effective public health tool to protect against severe COVID-19.

An important research question remains about the duration and breadth of T cell responses after natural infection or vaccination [19,20,21]. However, we found a clear correlation between anti-S anti-SARS-CoV-2 T cell immune responses and the height of antibody responses (Figure 5). Interestingly, a similar correlation was observed in convalescent patients with COVID-19, especially during the peak of antibody responses [22]. Therefore, as mentioned above, we do not recommend the study of cellular anti-SARS-CoV-2 responses as a routine laboratory test after vaccination.

In conclusion, our study provided clear evidence about the immunogenicity of BNT162b2 mRNA vaccination against SARS-CoV-2 while analyzing the intensity and dynamics of anti-SARS-CoV-2 immune responses. Our findings suggested that COVID-19 vaccination is an effective tool for protection against the consequences of the disease and allowed for the description of the most vulnerable groups who should be offered vaccines, including individuals aged 60 years or older, as well as individuals with chronic conditions.

## Figures and Tables

**Figure 1 vaccines-10-00316-f001:**
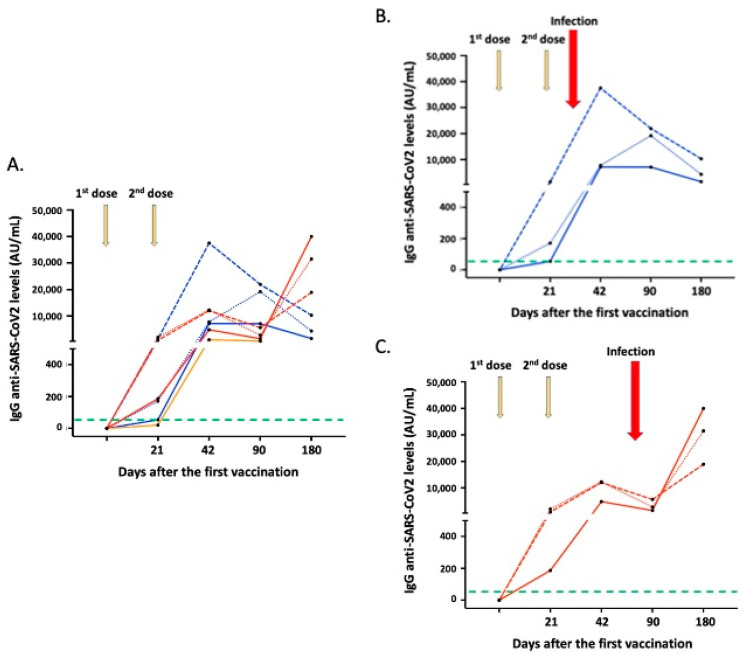
The dynamics of IgG anti-SARS-CoV-2 responses in vaccinated individuals infected after vaccination. (**A**) Line graphs of IgG responses in all participants. The orange line corresponds with IgG responses of the participant infected between the two vaccine doses; line (blue) graphs correspond with IgG responses of participants infected between 42–90 days post-vaccination (also presented separately in (**B**)); line (red) graphs correspond to IgG responses of participants infected between 90–180 days post-vaccination (also presented separately in (**C**)). The green dotted lines represent the cut-off of positive anti-SARS-CoV-2 IgG antibodies (50 AU/mL). Interestingly, participants infected between 90–180 days following the first vaccination displayed a further increase of anti-SARS-CoV-2 IgG levels on day 180 (**C**) in contrast to that observed in other study participants, since the infection seems to act as a booster vaccination.

**Figure 2 vaccines-10-00316-f002:**
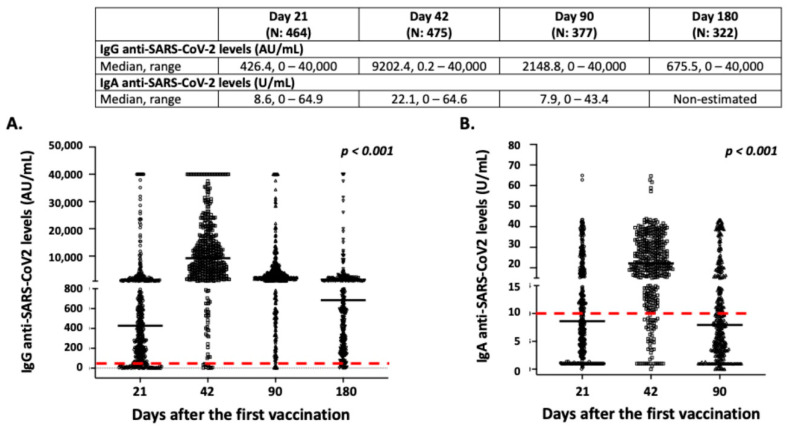
Intensity and dynamics of IgG (**A**) and IgA (**B**) anti-SARS-CoV-2 responses in the study participants. Black lines indicate median values, and red dotted lines represent the cut-off of positive anti-SARS-CoV-2 IgG (50 AU/mL) and IgA (10 U/mL) antibodies. Statistical significance refers to the Kruskal-Wallis H test.

**Figure 3 vaccines-10-00316-f003:**
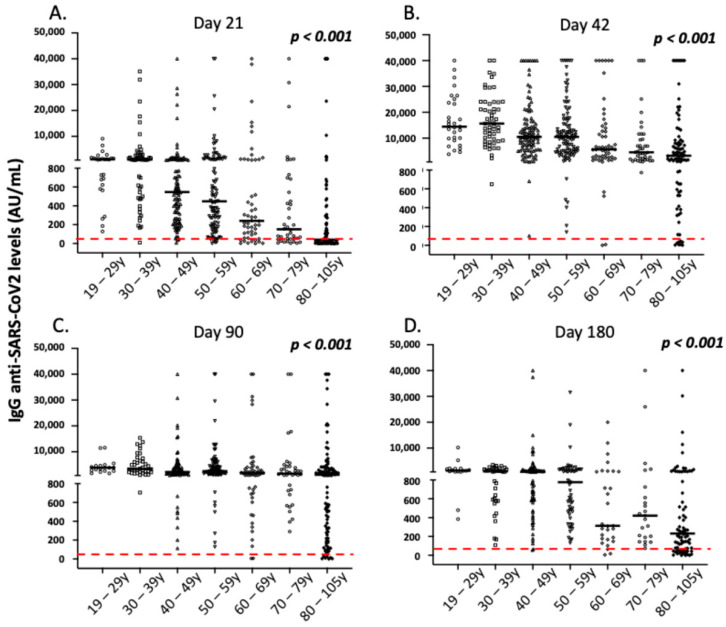
IgG anti-SARS-CoV-2 responses according to age groups in the study participants. (**A**) Day 21, (**B**) day 42, (**C**) day 90, and (**D**) day 180. Black lines indicate median values, and red dotted lines represent the cut-off of positive anti-SARS-CoV-2 IgG (50 AU/mL) antibodies. Statistical significance refers to the Kruskal-Wallis H test.

**Figure 4 vaccines-10-00316-f004:**
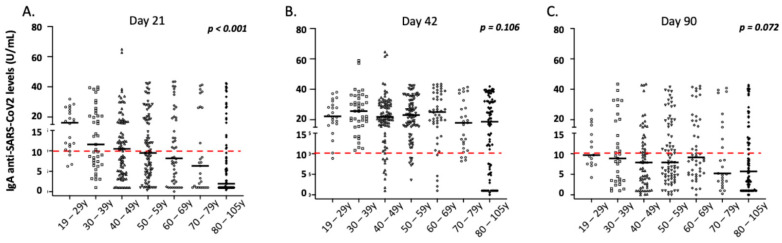
IgA anti-SARS-CoV-2 responses in the study participants according to age group. (**A**) Day 21, (**B**) day 42, and (**C**) day 90. Black lines indicate median values, and red dotted lines represent the cut-off of positive anti-SARS-CoV-2 IgA (10 U/mL) antibodies. Statistical significance refers to the Kruskal-Wallis H test.

**Figure 5 vaccines-10-00316-f005:**
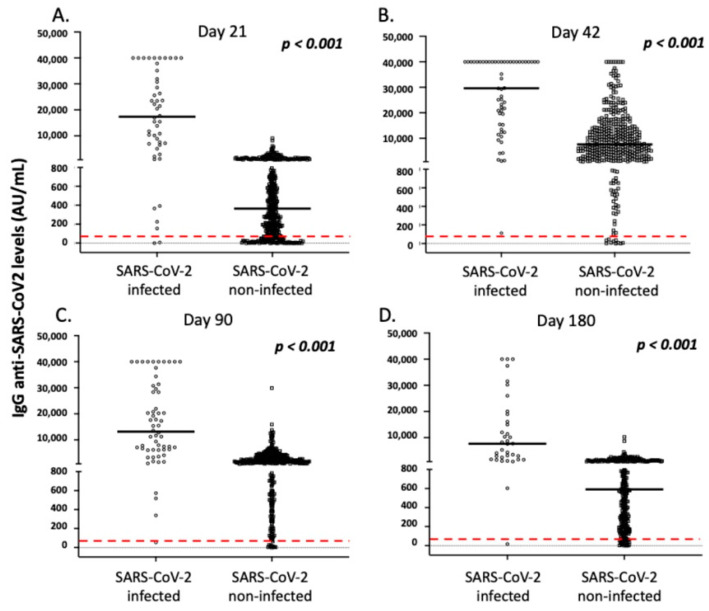
IgG anti-SARS-CoV-2 responses in the study participants according to history of COVID-19 infection (before and after vaccination). (**A**) Day 21, (**B**) day 42, (**C**) day 90, and (**D**) day 180. Black lines indicate median values, and red dotted lines represent the cut-off of positive anti-SARS-CoV-2 IgG (50 AU/mL) antibodies. Statistical significance refers to the Mann-Whitney U test.

**Figure 6 vaccines-10-00316-f006:**
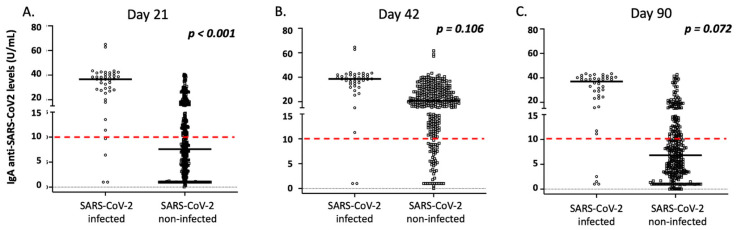
IgA anti-SARS-CoV-2 responses in the study participants according to history of COVID-19 infection (before and after vaccination). (**A**) Day 21, (**B**) day 42, and (**C**) day 90. Black lines indicate median values, and red dotted lines represent the cut-off of positive IgA (10 U/mL) antibodies. Statistical significance refers to the Mann-Whitney U test.

**Figure 7 vaccines-10-00316-f007:**
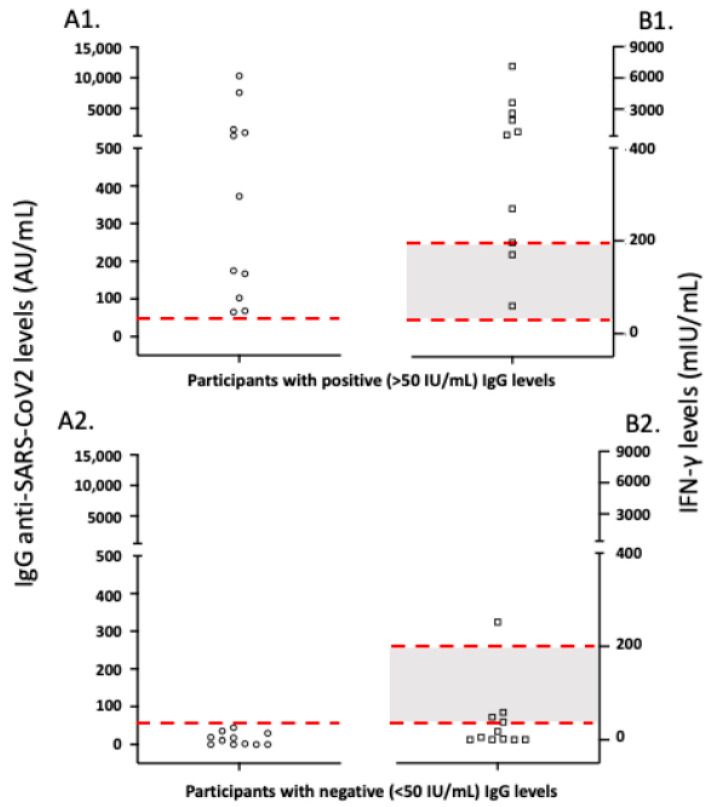
Correlation of humoral to cellular immune responses 6 months after anti-SARS-CoV-2 vaccination. The red dotted lines in parts (**A1**,**A2**) of the figure represent the cut-off of positive anti-SARS-CoV-2 IgG (50 AU/mL) antibodies, while in parts (**B1**,**B2**), they discriminate the negative and positive IFN-gamma levels; thus, the grey area represents the borderline region (100–200 mIU/mL) of anti-SARS-CoV-2 cellular immune responses.

**Table 1 vaccines-10-00316-t001:** Overview of demographic and clinical data of individuals (no. 511) enrolled in the study.

No.	511
Sex (male/female)	190/321
Age (median, range)	54 years, 19–105
Hypertension (*n*, %)	140, 27.4%
Diabetes mellitus (*n*, %)	68, 13.3%
Dyslipidemia (*n*, %)	89, 17.4%
Chronic heart disease (*n*, %) ^	74, 14.5%
Chronic respiratory disease (*n*, %) *	29, 5.7%
Stroke and transient ischemic attacks (*n*, %)	27, 5.3%
Thrombotic attacks (venous and/or arterial) (*n*, %)	11, 2.2%
Chronic liver disease (*n*, %) #	5, 1.0%
Chronic kidney disease (*n*, %) ^^	7, 1.4%
Thyroid disease (*n*, %) **	60, 11.7%
Autoimmune or autoinflammatory diseases (*n*, %) ##	41, 8.0%
Cancer (*n*, %) ^^^	27, 5.3%
Insomnia or psychiatric diseases (*n*, %) ***	126, 24.7%
Others ###	91, 17.8%
Comorbidities > 2 (*n*, %)	128, 25.0%
History of previous COVID-19 disease (*n*, %)	60, 11.7%

^ Chronic heart disease includes atrial fibrillation, arrhythmias, heart failure, and angina and myocardial infarction; * Chronic respiratory disease includes chronic obstructive pulmonary disease, chronic interstitial pulmonary disease, bronchiectasis, and asthma; # Chronic liver disease includes chronic HBV hepatitis, primary biliary cirrhosis, and alcohol-related liver disease; ^^ Chronic kidney disease includes chronic renal insufficiency and hydronephrosis; ** Thyroid disease includes Hashimoto disease with hypothyroidism, hypothyroidism (non-autoimmune), and hyperthyroidism; ## Autoimmune or autoinflammatory diseases includes lupus, rheumatoid arthritis, dermatomyositis, scleroderma, psoriasis and psoriatic arthritis, ancylosing spondylarthritis, uveitis, myasthenia gravis, Crohn disease, and ulcerative colitis; ^^^ Cancer includes patients with a history of gastric, breast, thyroid, or kidney cancer and hematologic malignancies (leukemia, Hodgkin lymphoma, Waldenstrom macroglobulinemia, and multiple myeloma); *** Insomnia or psychiatric diseases includes patients with insomnia, schizophrenia, depression, bipolar disorders, and psychosis; ### Others includes anemia, neutropenia, pernicious anemia, osteoarthritis, osteoporosis, history of epilepsy, Parkinson’s disease, prostate hyperplasia, hydrocephalus, Down syndrome, and history of kidney transplantation.

**Table 2 vaccines-10-00316-t002:** Adverse reactions after vaccination in study individuals.

	1st Dosage (*n*, %)	2nd Dosage (*n*, %)
Local pain	134, 36.9%	103, 28.5%
Fever	11, 2.4%	45, 9.7%
Myalgias	11, 3.0%	39, 10.8%
Fatigue	19, 5.2%	65, 18.0%
Headache	30, 8.3%	34, 9.4%
Flu-like symptoms	1, 0.3%	5, 1.4%
Others *	25, 6.9%	43, 11.9%

* Others includes individuals with chills (12), lymphadenitis (8), nausea (6), drowsiness (6), dizziness (5), weakness (4), numbness (4), vomiting (2), tachycardia (2), arthralgias (2), local edema (2), metal mouth sense (2), diarrhea (1), rash (1), mastodynia and mastalgia (1), skin irritation (1), vertigo (1), itching (1), loss of taste (1), atypical local spasm (1), flushing (1), ECG changes (1), and thorax pain (1).

**Table 3 vaccines-10-00316-t003:** IgG and IgA anti-SARS-CoV-2 responses after BNT162b2 mRNA vaccination in the study participants according to age group.

	Age 19–29 (N 29)	Age 30–39 (N 57)	Age 40–49 (N 109)	Age 50–59 (N 108)	Age 60–69 (N 53)	Age 70–79 (N 43)	Age 80–105 (N 112)	*p **
A. Day 21 after the first vaccination								
IgG levels (median, range)	1093.1, 127.7–9067.3	1008.2, 9.3–35,115.6	546.5, 7.0–40,000.0	448.4, 0.0–40,000.0	240.5, 0.0–40,000.0	151.2, 2.9–40,000.0	39.8, 0.0–40,000.0	<0.001
IgA levels (median, range)	16.2, 6.2–31.8	11.6, 1.0–39.9	10.5, 1.0–64.9	9.5, 0.4–42.6	8.2, 0.1–43.5	6.3, 1.0–41.3	1.9, 0.4–42.4	<0.001
B. Day 42 after the first vaccination								
IgG levels (median, range)	14,434.8, 3757.5–40,000.0	15,631.0, 649.3–40,000.0	10,485.9, 100.4–40,000.0	10,484.7, 140.5–40,000.0	5650.5, 0.2–40,000.0	4509.4, 773.4–40,000.0	3206.2, 1.6–40,000.0	<0.001
IgA levels (median, range)	22.2, 8.9–38.1	25.6, 10.8–58.9	21.8, 1.0–64.3	23.0, 3.6–42.6	25.0, 1.0–43.4	17.9, 8.2–41.4	18.6, 0.0–41.9	0.106
C. Day 90 after the first vaccination								
IgG levels (median, range)	3836.2, 1621.5–11,572.8	3376.9, 704.9–15,365.8	2180.1, 113.2–40,000.0	2491.6, 127.5–40,000.0	1713.7, 5.4–40,000.0	1550.3, 290.8–40,000.0	1092.7, 0.0–40,000.0	<0.001
IgA levels (median, range)	9.7, 4.2–26.2	8.9, 0.9–43.4	7.9, 0.0–43.9	7.9, 0.0–41.5	9.1, 0.1–42.1	5.2, 0.3–40.8	5.7, 0.0–42.7	0.072
D. Day 180 after the first vaccination								
IgG levels (median, range)	1303.1, 384.6–10,218.0	937.5, 108.2–3494.9	885.1, 55.2–40,000.0	776.1, 130.9–31,532.6	312.9, 5.0–20,003.0	420.6, 72.9–40,000.0	227.6, 0.0–40,000.0	<0.001
IgA levels	NE	NE	NE	NE	NE	NE	NE	

Abbreviations: N, number of participants; NE, non-estimated; * Statistical significance refers to the Kruskal-Wallis H test.

**Table 4 vaccines-10-00316-t004:** IgG (AU/mL) and IgA (U/mL) anti-SARS-CoV-2 responses after BNT162b2 mRNA vaccination according to COVID-19 infection.

	SARS-CoV-2 Infected	SARS-CoV-2 Non-Infected	*p*
A. Day 21 after the first vaccination			
N	48	416	
IgG levels (median, range)	17,353.0, 0.0–40.000	364.8, 0.0–9067.3	<0.001
IgA levels (median, range)	36.0, 1.0–64.9	7.7, 0.1–40.7	<0.001
B. Day 42 after the first vaccination			
N	56	419	
IgG levels (median, range)	29,646.0, 111.8–40,000.0	7593.0, 0.2–40,000.0	<0.001
IgA levels (median, range)	38.7, 1.0–64.3	20.8, 0.0–61.7	<0.001
C. Day 90 after the first vaccination			
N	58	349	
IgG levels (median, range)	13,163.7, 54.9–40,000.0	6788.0, 0.0–29,894.9	<0.001
IgA levels (median, range)	37.0, 1.0–41.4	6.9, 0.0–42.7	<0.001
D. Day 180 after the first vaccination			
N	36	286	
IgG levels (median, range)	7646.2, 17.3–40,000.0	589.8, 0.0–10,218.0	<0.001
IgA levels (U/mL)	NE	NE	

Abbreviations: N, number of participants; NE, non-estimated. The number of participants with SARS-CoV-2 infections over time is presented in detail in the Results section.

**Table 5 vaccines-10-00316-t005:** Adverse side effects and intensity of IgG responses on day 42.

Parameter	N, Median IgG Levels (mg/dL) (IQR)	*p*, Univariate Analysis	*p*, Multivariate Analysis	Coefficient, 95% CI
Sex (male/female)	Male: 181, 8902 (16,190) Female: 294, 9462 (11,759)	0.789	-	
Age	ρ (rho) = −0.392	<0.001	0.026	−126.96, −238.69–−15.24
Comorbidities (≤2 vs. ≥3)	≥3: 111, 4828.8 (11,196.1) No: 364, 10,295.8 (12,503.9)	<0.001	0.518	−554.95, −2243.57–1133.66
Local pain	Yes: 102, 11,441.6 (12,279.0) No: 255, 9745.2 (12,705.1)	0.358	0.667	−563.54, −3146.07–2018.99
Fever	Yes: 45, 17,225.8 (17,149.2) No: 414, 7772.5 (11,209.8)	<0.001	0.004	5865.08, 1940.54–9789.61
Myalgias	Yes: 39, 17,501.4 (12,504.7) No: 318, 9308.1 (11,434.3)	<0.001	0.038	3911.15, 215.17–7607.14
Fatigue	Yes: 64, 14,209.7 (14,919.9) No: 293, 9050.9 (11,839.9)	<0.001	0.758	517.39, −2794.27–3829.05
Headache	Yes: 34, 10,152.8 (11,980.4) No: 323, 10,094.5 (12,170.3)	0.756	-	
Flu-like symptoms	Yes: 5, 15,764.4 (29,612.3) No: 352, 10,034.6 (12,538.6)	0.109	0.135	7443.16, −2325.49–17,211.82
Others *	Yes: 42, 11,899.3 (15,455.4) No: 315, 9640.2 (11,653.1)	0.119	0.798	446.68, −2993.8–3887.17

Abbreviations: IQR, interquartile range; 95% CI, 95% confidence interval; ρ (rho), Spearman’s rank correlation coefficient. The comorbidities are as presented in detail in Table 1, while the side effects are those that emerged after the second dose. * Others (adverse side effects) are described in detail in Table 2.

## Data Availability

The data that support the findings of this study are available from the corresponding author upon request.

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
