# Peer review of "Intensity and Dynamics of Anti-SARS-CoV-2 Immune Responses after BNT162b2 mRNA Vaccination: Implications for Public Health Vaccination Strategies"

_vaccines, 2022, doi:10.3390/vaccines10020316_

Round 1

Reviewer 1 Report

The manuscript Vaccines-1574781 from the field of public health / COVID-19 is an observational study reporting the IgG, IgA, and T-cell (IFN gamma assay) responses upon mRNA vaccination in a population sample of N=509 (double vaccinated)

The manuscript is relevant to the aims and scopes of Vaccines, is written in a competent language and structure, has a proper and reproducible methodological description, and has an appropriate statistical overworking.

MAJOR COMMENTS

The reviewer shall criticize that the figure and caption quality are inferior to the otherwise high manuscript quality and should be improved before publication according to the following guidelines:

C001 – Please increase the font size to an 8-10pt minimum and keep it uniform.

C002 - Add units on Y-axis (IU/ml) and X-axis (days after the first vaccination).

C003 - Add a subpanel with color/shape explanation in the figure for clarity, using 1 as an example. In Figure 1a, for example, it is unclear what other colors (apart from orange) represent and the difference between continuous and dotted lines. Even if this information is somewhere encrypted in the text, it is an essential element that should be obvious even to the furious reader.

C004 –Graphical representation of parametric data should provide a clue on a dataset's center (average/median) and dispersion (SD, SEM …). This is not the case in figure 1. Please upgrade using, for example, parametric plotting from 2,3.  

C005 - Figure 2: Figure 2A is not consistent with figure 1A. In figure 1A, there is no decrease in IgG after 180 days; in Figure 2A, the authors describe a significant reduction. Please rectify this conflict.  

C006 - Add N, P, and statistical test in all figure captions. See result reporting 4

Minor comments

  • N of double vaccinated =509, not 511. Please clearly phrase 
  • L58: 30@g, please correct
  • L186: A point is missing after "B"

Author Response

“The reviewer shall criticize that the figure and caption quality are inferior to the otherwise high manuscript quality and should be improved before publication according to the following guidelines:

Response: We modified the figures appropriately. In order to improve their quality we separated the data from two figures (thus we added 2 more figures and 2 Tables in order to present our results according to the reviewer’s recommendation)

C001 – Please increase the font size to an 8-10pt minimum and keep it uniform.

Response: The Figure was prepared accordingly

C002 - Add units on Y-axis (IU/ml) and X-axis (days after the first vaccination).

Response: The Figure was prepared accordingly

C003 - Add a subpanel with color/shape explanation in the figure for clarity, using 1 as an example. In Figure 1a, for example, it is unclear what other colors (apart from orange) represent and the difference between continuous and dotted lines. Even if this information is somewhere encrypted in the text, it is an essential element that should be obvious even to the furious reader.

Response: Each line corresponds to a separate participant. We also modified the figure legend in order to present our data more clearly

C004 –Graphical representation of parametric data should provide a clue on a dataset's center (average/median) and dispersion (SD, SEM …). This is not the case in figure 1. Please upgrade using, for example, parametric plotting from 2,3. 

Response: As mentioned above, we added two more Tables and modified the Figures in order to present our data according to reviewer’s recommendation

C005 - Figure 2: Figure 2A is not consistent with figure 1A. In figure 1A, there is no decrease in IgG after 180 days; in Figure 2A, the authors describe a significant reduction. Please rectify this conflict. 

Response: We consider that after the modification of Figure 1 legend the data are clear demonstrating that in 3 participants who infected by SARS-CoV-2 between 90-180 days after the first vaccination, the infection seemed to act as a booster vaccination and this is the reason that they have higher IgG levels on day 180

C006 - Add N, P, and statistical test in all figure captions. See result reporting 4

Response: p and statistical test are included in Figure captions appropriately, and the N (number of participants for each parameter) have been added in detail in the new Tables (3 and 4) which added in the new version of our manuscript. We consider that this option is better, since in Tables we had the option to present all data in detail.

Minor comments

N of double vaccinated =509, not 511. Please clearly phrase

Response: Actually the double vaccinated participants were 509, and this information was also clarified in abstract and everywhere into the text

L58: 30@g, please correct

Response: We corrected it appropriately (it is a symbol, the “mi” in Greek terminology, namely “microgrammar”)

L186: A point is missing after "B"”

Response: The Figure legend was modified

We thank the Reviewer for his/her positive and accurate comments for our study. All his/her recommendations were appropriately addressed, and the quality of our manuscript was substantially improved.

Reviewer 2 Report

Thank you for the opportunity of reviewing this paper. The research is interesting, results add important evidence and the paper is well written.

Please, find here my suggestions/comments.

First of all, anti-SARS-CoV-2 vaccine (e.g., lines 46, 70, 79, and others) must be changed to COVID-19 vaccine. Indeed, the protection offered by the current developed and authorized vaccine is only against the consequences of the disease, not against infection sustained by SARS-CoV-2 (as observed with the breakthrough infections). This is a very relevant difference that must be addressed before publication of this interesting paper.

Line 58, amend “two 30-@g doses“

Line 76, remove “(60)”

Please, compare your results with important evidence in the field; for example, let me suggest the from a comprehensive ongoing longitudinal study (VASCO project) on the response to BNT162b2 COVID-19 vaccine (of which I am NOT an author):

- Ponticelli D, Madotto F, Conti S, Antonazzo IC, Vitale A, Della Ragione G, Romano ML, Borrelli M, Schiavone B, Polosa R, Ferrara P, Mantovani LG.

Response to BNT162b2 mRNA COVID-19 vaccine among healthcare workers in Italy: a 3-month follow-up. Intern Emerg Med. 2021 Oct 12:1-6. doi: 10.1007/s11739-021-02857-y

- Ponticelli D, Antonazzo IC, Caci G, Vitale A, Della Ragione G, Romano ML, Borrelli M, Schiavone B, Polosa R, Ferrara P. Dynamics of antibody response to BNT162b2 mRNA COVID-19 vaccine after 6 months. J Travel Med. 2021;28(8):taab173. doi: 10.1093/jtm/taab173.

Please, better discuss the role of previous infection as immune priming (the suggested references are both useful also for this point).

Line 357. “We suggest that individuals older than 60 years of age represent the most important target group for intensification of vaccination” should be rephrased to “Our findings suggest that COVID-19 vaccination is an effective tool for the protection against the consequences of the disease, allowing to describe the most vulnerable groups that should be offered with vaccines, including individuals aged 60 years or older and people with chronic conditions.

Further perspectives:

I don’t know how the study database was assembled. A plenty of factors have proven to be an effect on the responses to COVID-19 vaccines, including smoking. Is it possible to investigate the impact of smoking on humoral re response?

Author Response

“Thank you for the opportunity of reviewing this paper. The research is interesting, results add important evidence and the paper is well written.

Please, find here my suggestions/comments.

First of all, anti-SARS-CoV-2 vaccine (e.g., lines 46, 70, 79, and others) must be changed to COVID-19 vaccine. Indeed, the protection offered by the current developed and authorized vaccine is only against the consequences of the disease, not against infection sustained by SARS-CoV-2 (as observed with the breakthrough infections). This is a very relevant difference that must be addressed before publication of this interesting paper.

Response: We strongly agree and we have corrected the terminology throughout the manuscript

Line 58, amend “two 30-@g doses“

Response: We corrected it appropriately (it is a symbol, the “mi” in Greek terminology, namely “microgrammar”)

Line 76, remove “(60)”

Response: It is removed

Please, compare your results with important evidence in the field; for example, let me suggest the from a comprehensive ongoing longitudinal study (VASCO project) on the response to BNT162b2 COVID-19 vaccine (of which I am NOT an author):

- Ponticelli D, Madotto F, Conti S, Antonazzo IC, Vitale A, Della Ragione G, Romano ML, Borrelli M, Schiavone B, Polosa R, Ferrara P, Mantovani LG. Response to BNT162b2 mRNA COVID-19 vaccine among healthcare workers in Italy: a 3-month follow-up. Intern Emerg Med. 2021 Oct 12:1-6. doi: 10.1007/s11739-021-02857-y

- Ponticelli D, Antonazzo IC, Caci G, Vitale A, Della Ragione G, Romano ML, Borrelli M, Schiavone B, Polosa R, Ferrara P. Dynamics of antibody response to BNT162b2 mRNA COVID-19 vaccine after 6 months. J Travel Med. 2021;28(8):taab173. doi: 10.1093/jtm/taab173.

Please, better discuss the role of previous infection as immune priming (the suggested references are both useful also for this point).

Response: We referred the first study (lines 334-342), where similar results with a waning of IgG responses in vaccinated participants was observed over time. Moreover, we also discussed the discrepancy between our results and the Italian study, since we observed a clear correlation of antibody responses with age, that was not demonstrated in the Italian study. As we commented: “The discrepancy with our results may be attributed to the fact that Ponticelli et al., analyzed the immune responses of healthcare workers after vaccination, namely they did not include elderly individuals; obviously this is an advantage of our study, where we evaluated the immune responses after vaccination on a wider spectrum of age”. The role of COVID-19 infection is also discussed in lines 343-350, and we have also added a comment in the Caption of Figure 1, where “the participants infected between 90-180 days after the first vaccination displayed a further increase of anti-SARS-CoV-2 IgG levels on day 180 (C), in contrast to that observed to other study participants, since the infection seems to act as a booster vaccination”.

Line 357. “We suggest that individuals older than 60 years of age represent the most important target group for intensification of vaccination” should be rephrased to “Our findings suggest that COVID-19 vaccination is an effective tool for the protection against the consequences of the disease, allowing to describe the most vulnerable groups that should be offered with vaccines, including individuals aged 60 years or older and people with chronic conditions.

Response: We are grateful to the reviewer for this suggestion that we have included in the revised version of our manuscript, according to his/her recommendation.  

Further perspectives:

I don’t know how the study database was assembled. A plenty of factors have proven to be an effect on the responses to COVID-19 vaccines, including smoking. Is it possible to investigate the impact of smoking on humoral response?:

Response: We strongly agree, but unfortunately, we do not have this data for all participants.

Reviewer 3 Report

The article entitled “Intensity and dynamics of anti-SARS-Cov-2 immune responses after BNT162b2 mRNA vaccination: Implications for public health vaccination strategies” is a study about the intensity and dynamic (on days 21, 42, 90 and 180) of the humoral and cellular immune response after BNT162b2 mRNA vaccination by measuring serum IgG and IgA antibodies and SARS-CoV-2 IFN-gamma release.

The results of this study showed:

  1. age over 60 years is associated with lower IgG;
  2. previous COVID-19 infection and adverse side reactions were associated with higher IgG;
  3. High IgG levels at 3-6 months
  4. Low IgA levels at 6 months

In addition it was reported that there was a correlation between the serum antibodies levels and the T-cellular response and, therefore, the study of cellular responses is not indicated as a routine laboratory test after vaccination.

I think that these informations are useful for the scientific community and the readers of this journal.

The article is well-written in terms of purpose and methods.

I think that this article is suitable for publication in its current version.

Author Response

“… I think that this article is suitable for publication in its current version”.

 Response: We are grateful to the reviewer for his/her comments.